# The Rural Fires of 2017 and Their Influences on Water Quality: An Assessment of Causes and Effects

**DOI:** 10.3390/ijerph20010032

**Published:** 2022-12-20

**Authors:** Mário David Sequeira, Ana Castilho, Alexandre Oliveira Tavares, Pedro Dinis

**Affiliations:** 1Department of Earth Sciences, Centre for Social Studies, University of Coimbra, 3030-790 Coimbra, Portugal; 2Department of Earth Sciences, Geosciences Centre, University of Coimbra, 3030-790 Coimbra, Portugal; 3Department of Earth Sciences, MARE–Marine and Environmental Sciences Centre/ARNET-Aquatic Research Network, University of Coimbra, 3030-790 Coimbra, Portugal

**Keywords:** surface water quality, rural fires, land-use, physical-chemical parameters, water influencing factors

## Abstract

As water is facing increasing pressures from population and economic growth and climate change, it becomes imperative to promote the protection, restoration and management of this resource and its watersheds. Since water quality depends on multiple factors both natural and anthropic, it is not easy to establish their influences. After the October 2017 fires that affected almost 30% of the Mondego hydrographic basin in Central Portugal, 10 catchments were selected for periodic physical-chemical monitoring. These monitoring campaigns started one month after the fires and lasted for two hydrological years, measuring the electric conductivity (EC), pH, dissolved oxygen (DO), turbidity (Turb), alkalinity (Alk), major and minor ions, and trace elements. The obtained data were then statistically analysed alongside the geomorphological characteristics of each catchment coupled with features of land-use and occupation. From the results, it was possible to establish that fire-affected artificial areas, through the atmospheric deposition and surface runoff of combustion products, had the most impact on surface water quality, increasing As, K^−^, Ca^2+^, Mg^2+^, NO_3_^−^, SO_4_^2−^ and Sr, and consequently increasing electrical conductivity. Agricultural land-use seems to play a major influence in raising the water’s EC, Cl, K^−^ and Na^2+^. Regarding natural factors such as catchment geology, it was found that the extent of igneous exposures influences As, and the carbonate sedimentary units are a source of Ca^2+^ and HCO_3_^2−^ concentrations and impose an increase in alkalinity. Rainfall seems, in the short term, to increase the water concentration in Al and NO_3_^−^, while also raising turbidity due to sediments dragged by surface runoff. While, in the long-term, rainfall reduces the concentrations of elements in surface water and approximates the water’s pH to rainfall features.

## 1. Introduction

The protection, restoration and management of water and soil are some of the key objectives of the United Nation’s Sustainable Goals for 2030 [1]. As it is witnessed, the increasing devastating impacts of climate change [2] and population and economic growth [3,4], has place water resources at a high risk.

In the last two decades, there has been increased concern regarding the quality of surface waters, as rivers are a water resource for domestic, industrial and agricultural purposes, and play a fundamental role in the hydrological and biogeochemical cycles [5]. River systems have been negatively affected by the interaction of multiple factors [6]. Among these, changes in land use [7] and fires [8] are proven to critically affect river water quality at spatial and temporal scales. Land use and water quality studies have emphasized anthropic activities (e.g., agriculture and urbanization) as a major source of non-point source pollution [9], predominantly in increasing nutrients (i.e., nitrogen and phosphorus) [10,11,12,13,14,15,16,17], total dissolved solids [12,17] and turbidity [12], while diminishing dissolved oxygen (DO) [15,17]. It has also been witnessed that rural fires can become a major environmental issue for water quality [18,19,20,21]. Following the fires, especially after high precipitation periods, the erosion rates increase [20] and the runoff transports sediment and ashes through the burnt slopes [22,23], leading to increases in total suspended solids [8,21,24] and nutrients [21,25].

Despite the fact that these relationships have received more attention in the scientific literature in recent years, little attention has been given to the changes of trace elements in surface water in relation to land use and/or fires [26,27,28,29]. Furthermore, most studies tend to focus on experimental conditions with low-order streams, a high variety of land uses and short observable periods [30].

Bearing in mind that understanding the factors that influence surface water quality are important to establish an effective water management system [6], this study aims to explore the relationship of specific drivers with surface water quality. More specifically, this study aims at ascertaining the influence of land use and fires, considering their interplay with other environmental features (i.e., geology and precipitation) on water quality. Moreover, this study aims to discuss whether the founded water parameters, particularly those related to fires, may have an impact in public health, specifically regarding the water’s safe consumption. For this, the results of nine monitoring campaigns in ten catchments spanning two years were statistically analysed in order to determine the significant factors influencing water quality in the Mondego hydrological basin of Central Portugal (Figure 1).

## 2. Study Area

The Mondego River drainage basin in the Central Region of Portugal was significantly affected by the October 2017 fires, which affected around 30% of its area (Figure 2). The Mondego River Hydrological Basin is composed of the Mondego River, its 6 tributary rivers (Alva, Ceira, Dão, Pranto, Arunca and Ega) as well as a myriad of small watercourses. All of these rivers, as well as the monitored courses, are perennial.

In the region are cropped out Paleozoic granitoids, mainly in its northern and eastern sectors; Precambrian-Paleozoic meta-sedimentary units, mainly to the south; and sedimentary units in small continental basins and river valleys [31]. To the west lies a Meso-Cenozoic sedimentary succession with siliciclastic and carbonate units (Table 1). The monitored catchments (MC) also have high variable morphologies, as the southernmost present higher slopes contrasting to flatter catchments to the north and particularly to the east (Table 1).

In relation to the land use and occupation (LUO), most of the south and east of the study area is occupied by forests of *Quercus*, *Pinus pinaster* and Eucalyptus. The majority of the downstream catchments have wider artificial areas (Table 1). Although most catchments present an equilibrium between forest, shrub and/or herbaceous vegetation, and agricultural land use, some MCs can have considerable artificial areas, such as MC10 (Table 1).

The study area is included in the class of temperate dry and hot summer (Csb) according to the Köppen-Geiger classifications. The climate normal (1971–2000) highlights July and August as the driest months (<18 mm), and December as the wettest (>120 mm) [28].

## 3. Methods

In order to distinguish water parameters that were influenced by the fires from other factors, a statistical analysis was conducted. The water’s physical and chemical analysis served as the dependent variables, while the chosen independent variables were the catchments’ characteristics and the precipitation daily (P) and of the last 5 (P5) and 10 (P10) days before each campaign. As most of the intended study elements tend to have a conservative nature at the catchment scale, this was the scale used to determine the independent variables [32]. All precipitation data were collected from weather stations close to the catchments, with continuous precipitation data for the study period (Coimbra: 40°12′0″ N, 8°26′60″ W and Viseu: 40°39′60″ N, 7°54′0″ W).

### 3.1. Measurements of the Catchments Characteristics

Vector-based datasets were collected for altimetry [33], geology [34], LUO [35] and the Copernicus Emergency Management Service (EMS)—Mapping (Copernicus). The criteria for choosing the datasets were the availability and resolution proximity between them. All spatial analyses were conducted with the software ArcGIS, version 10.7.1 (Esri: Redlands, CA, USA).

The average slope (AvSlp) was determined through the Zonal Statistics toolset from altimetry data. The geological data was simplified into clastic sedimentary (SClt), carbonate sedimentary (SCrb), igneous (Ign) and metamorphic (Mtm) geology categories. The CORINE LUO data, which was from 2018, had to be updated for the studied periods. Google Earth CNES/Airbus imagery from the studied catchments were imported to ArcGIS, as were each shapefile to the intended timeframe [36]. The CORINE LUO shapefiles were combined afterwards into artificial (Art) and agricultural (Agr) areas. Also, forest (For) and shrub/herbaceous vegetation (Shv) were created by combining forest and shrub and/or herbaceous vegetation association sub-classes. The LUO was overlapped with the Copernicus EMS—Mapping to define how much each LUO was impacted by fire, defining the fire-affected areas (A). All areas were calculated for each basin through the Calculate Geometry function of ArcGIS.

### 3.2. Water Sampling and Analysis

The surface-water monitoring campaigns were conducted during two hydrological years (2017/18 and 2018/19). During the first hydrological year, 7 campaigns were conducted from November to June with a periodicity of around 36 days. While in the hydrological year of 2018/19, two campaigns were conducted, one in the rainy season (April) and the other in the dry season (September). In each campaign, the waters’ electrical conductivity (EC), pH, DO, turbidity (Turb) and alkalinity (Alk) were determined in situ [29]. The waters’ EC, pH and DO were measured using the WTW multiparameter probe 340i, and the turbidity was measured using the HANNA HI 93102 Turbidity Portable Meter. To determine the water’s alkalinity, the ISO 9963-1 standard was used. Additionally, water samples were collected for the assessment of major and minor ions (Br^−^, Cl^−^, NO_3_^−^, PO_4_^3−^, SO_4_^2−^, Ca^2+^, K^−^, Mg^2+^ and Na^2+^) and metals and metalloids (Al, As, Ba, Fe, Mn, Ni, Pb, Sr and Zn). The surface water was not analysed for dissolved organic carbon or total organic carbon, since in the same conditions all values have been below the detection limits [28]. These analyses were conducted by an accredited laboratory (the Itecons—Instituto de Investigação e Desenvolvimento Tecnológico para a Construção, Energia, Ambiente e Sustentabilidade). The analyses of the major anions were conducted in a timeframe within 24 h of the sampling. The techniques used; their precision, detection and quantification limits; as well as the used standards are in Table 2.

### 3.3. Statistical Analysis

The statistical analysis was conducted using IBM SPSS Statistics version 26. All water parameters below detection levels were replaced by half of their limit values [37]. All variables were verified for outliers using boxplots and the interquartile (IQR) method. After removing all outliers, the covariance between variables were evaluated with the Pearson correlation coefficient, allowing for the defining of statistically significant relationships between the independent variables and the water parameters (*p* < 0.01 and *p* < 0.05). This procedure was adopted to select the most useful set of independent variables for the regression analysis. Violations of the assumptions of the Pearson correlation coefficient (e.g., level of measurement, related pairs, inexistence of outliers) were assessed beforehand. A violation of the assumptions was detected for PO_4_^3−^.

A regression analysis model was constructed from the significant independent variables to calculate the influences on the water parameters. Using a stepwise regression method, less significant independent variables were removed from the model. For each tested model, violations of the assumptions were assessed beforehand. To ensure that the independent variables were truly independent, the tolerance and variance inflation factor (VIF) were also determined and considered in the construction of the model.

## 4. Results

From the analytical results, it was noticed that almost all of the PO_4_^3−^ concentrations were below the detection limit; after the exclusion of its outliers, it was observable that all the remaining values of PO_4_^3−^ were bellow detection levels, making this variable constant in all accounted samples. This meant that for this parameter, it would not be possible to establish any type of correlations. The statistical representation of the results after the removal of the outliers are presented in Table 3.

### 4.1. Pearson Correlation

The Pearson correlation results are presented in Table 4. Based on the criteria presented in Section 3.3, and given the absence of significant correlations with Ni and Zn, these elements were not considered for the regression analysis.

### 4.2. Regression Analysis between Possible Influences and Water Parameters

The results show that water parameters cannot be well described by single factors, although they can be fairly predicted with two or more factors (Table 5). The prominent examples of the water parameters that were well projected using the independent variables were As, K, Ca^2+^ and NO_3_^−^.

Multicollinearity was observed between some independent variables (i.e., Art * SCrb, R = 0.998, *p* < 0.001 and AAgr * AArt, R = 0.757, *p* < 0.001), requiring the removal of less predictive variables, which in this case were the Art and the AAgr areas. The tolerance, VIF and normality plots of the residuals for the models are describe in the Appendix A.

## 5. Discussion

Data exploitation using multiple regression analysis had the advantage of categorizing the studied independent variables and grading them in terms of influence. Even in cases where the obtained R^2^ for the dependent variables are low, such as for Turb, HCO_3_^2−^, Br, Fe, Mn and Pb, resulting in a less robust prediction model, they can still display which independent variables exert a stronger influence.

### 5.1. Rainfall

Rainfall has two particularly distinguishable effects: (a) it transports sediments and chemical compounds to watercourses through runoff [38], raising the concentrations of some ions; and (b) it acts as a dilution agent, reducing these same concentrations [39]. The results seem to suggest that the steadiness of the rainfall affects the dependent variables differently. Short term rainfall, represented by P, was the most influencing factor in increasing Al, while P5 was the most influencing variable in reducing DO and Pb. Continuous rainfall, expressed through P10, is the main factor responsible for the increases in the water’s turbidity; the concentration of Al and NO_3_^−^, nevertheless, reduces the water’s pH to levels comparable to the rainfall [40].

### 5.2. Geology

The proportion of granitoids in the catchment seems to contribute to increases in the As content in the water, as this element has its origin in a wide variety of minerals present in this type of geology [41]. However, one possible reason for this type of geology not having a stronger influence might be related to As having a low environment circulation [42]. The concentrations of Ca^2+^ and Fe in surface waters also seem to be influenced by the spatial representation of granitoids, but to a lower level. The metamorphic areas had a minor negative influence on the quantity of major and trace constituents (i.e., HCO_3_^2−^, K^−^, NO_3_^−^, SO_4_^2−^, Ba, Sr and Mn). This low mineralization might be related to the small interaction between the surface runoff and the metamorphic units. The high permeability of the clastic sedimentary units could be the cause for this type of geology having a residual impact on the surface water’s mineralization. The presence of sedimentary units plays a strong influence on Ca^2+^ and HCO_3_^2−^ concentrations, also increasing alkalinity. This is particularly visible in the MC10, which is mostly dominated by carbonate geology.

### 5.3. Land-Use and Occupation

Natural areas, both forest and shrub or herbaceous, appear to have a minor influence on the surface water chemistry. Catchments dominated by natural areas, particularly if thriving in forest and grassland, tend to have small water mineralization as they reduce surface runoff and soil erosion, decreasing the release of both particulate and dissolved pollutants in watercourses after precipitation events [43]. This, in turn, has the potential of increasing water quality in catchments with high natural areas [44]. Also, some vegetal species have a tendency to bioaccumulate particular chemical elements—for example the *Smilax aspera*, commonly found in the Mediterranean shrub lands, which bioaccumulates Ba [45].

Contrariwise, major ions (such as Cl^−^, K^−^ and Na^2+^) in surface water appear to be partially derived from agricultural activities. Fertilizers (i.e., KCl and NaNO_3_) and herbicides (i.e., NaClO_3_) can contribute to the increases in these elements in surface water, particularly in cases of excessive soil remobilization and irrigation [41].

Even where urban areas occupy small percentages of the catchment, its impacts on water quality can be high [30] either due to the discharge of untreated wastewater or the lack of permeable surfaces [13]. However, in this particular case, the majority of the monitored watersheds had residual artificial land-use, of which the majority corresponded to a discontinuous urban fabric and/or presented high collinearity with the carbonate sedimentary extent of the catchments, which might explain why it might seem like this variable has no influence on water quality.

### 5.4. Fire

Observing the fire-affected land-use areas requires some cautiousness since the designation “fire-affected” was used as an umbrella term for areas differently affected by fires. The data used did not allow for the distinguishing of the severity of the fire on the affected areas.

The extension of the forest burnt area appears to correlate with the water’s DO. Eutrophication processes tend to occur in the downstream areas of fire affected areas [46]. However, it was not observable in this particular case, being noteworthy that eutrophication requires a combination of factors such as the increasing temperatures and excessive inputs of nutrients (particularly PO_4_^3−^) in surface water, creating disproportionate algae blooms [47].

It is well known that anthropogenic activities can enhance concentrations of As in soils and waters [48] due to the use of As-based pesticides, herbicides and wood treatments [49]. Fires can also influence the availability of As in the environment, especially through atmospheric deposition [8,50]. In this particular study, the fire-affected area was the variable in the increased concentration of As in the surface water. The combustion of organic matter tends to released high quantities of nitrogen species (i.e., NO_3_^−^) and SO_4_^2−^ [16,25,29,51], while the latter can also be released from the ash deposits and the soil’s organic matter [8]. After the fire, the remaining ashes contain, depending of their nature, a diversity of elements such as Ca^2+^, Mg^2+^ and K^−^ in Pinus ashes [52,53], that become mobilizable. Finally, after a major fire event, as a consequence of the removal of vegetation, increases in runoff are observable [54]. The newfound erosion capacity from the water, and the increasing transport of all the aforementioned elements and sediments to the watercourses, leads to the contamination of water supplies and increases in the water’s EC [8,52].

When comparing the monitored parameters that appear to be influenced by fires with the Guidelines for Drinking-water Quality [55], only As presented values above the provisional guideline value of 10.00 µg/L. This occurred two months after the fire in MC2, when the As concentration in the surface water reached a value of 16.00 µg/L. Still, in a catchment with similar characteristics to the previous (MC2), there was an exceeding concentration after the recovery period (12.65 µg/L) in September of 2019. During the creation of the As model, the agricultural land-use had to be removed for the high collinearity with the igneous extension of the catchments, which led to not all independent variables having accounted for this element. As was mentioned previously, agricultural activity can be responsible for the presence of As in surface water, although in this study it wasn’t possible to confirm this premise.

## 6. Conclusions

The multiple regression analysis allows for the better understanding of the relationship between water’s physical-chemical parameters and its influences. Nevertheless, further studies should circumvent choosing catchments than can have high intercorrelations among its characteristics.

Rainfall in the short term introduces sediments and chemical compounds in surface water (affecting Al, NO_3_^−^ and Turb), while in the long term acts as a dilution agent, reducing element concentrations and decreasing pH to a rainfall equivalent.

Geology had very different effects on surface water, depending on rock type. Metamorphic and clastic sedimentary units tended to have lower water mineralization. Carbonate geology contributed the raising of the water’s Ca^2+^, HCO_3_^2−^ and alkalinity. Arsenic seems to be partially sourced from igneous rocks.

Natural areas such as forest, shrub and herbaceous areas did not seem to significantly influence surface water parameters. The use of chemical compounds, soil remobilization and irrigation by agricultural activities seemed to increase the water’s Cl^−^, K^−^, Na^2+^ and electrical conductivity. Artificial areas did not seem to influence water chemistry, either due to low representation in most catchments, or multicollinearity issues with sedimentary areas.

Fire-affected artificial areas seemed to be the most impactful factor on the surface water quality. Whether or not through atmospheric deposition and surface runoff of burnt organic matter and ash compounds, it appeared responsible for increasing concentrations of As, K^−^, Ca^2+^, Mg^2+^, NO_3_^−^, SO_4_^2−^ and Sr, and consequently for electrical conductivity. From all of the water parameters influenced by fire activity, As can exceed guideline values, although an individually high As concentration after the recovery period indicated that other source of As must also have been involved.

## Figures and Tables

**Figure 1 ijerph-20-00032-f001:**
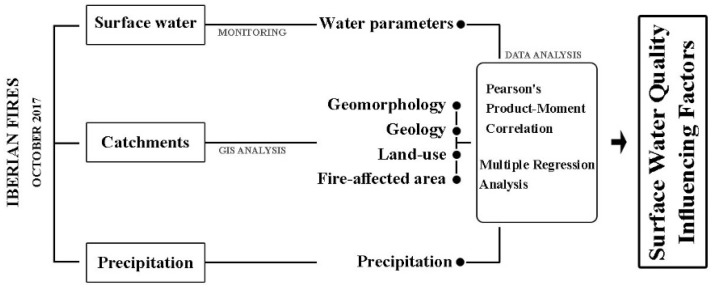
Conceptual model for determination of influences on water quality.

**Figure 2 ijerph-20-00032-f002:**
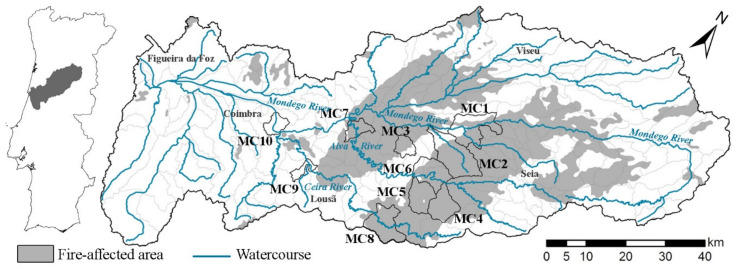
Study area and the location of the monitored catchments.

**Table 1 ijerph-20-00032-t001:** Monitored catchments areas, their average slope (AvSlp), area of clastic sedimentary (SClt), carbonate sedimentary (SCrb), igneous (Ign), metamorphic (Mtm) geology, area of artificial (Art), agricultural (Agr), forest (For), shrub and/or herbaceous vegetation (Shv) LUO and fire-affected counterparts (A). The AvSlp is in degrees and all areas are in percentage of the watershed.

	Watercourse	Geomorphology					LUO				Fire-Affected LUO			
Ref		AvSlp	SClt	SCrb	Ign	Mtm	Art	Agr	For	Shv	AArt	AAgr	AFor	AShv
MC1	Mondego	5.1	3.9	0.0	96.1	0.0	0.8	31.5	14.0	53.7	0.2	16.3	8.1	38.7
MC2	Cavalos	4.5	9.8	0.0	87.4	2.8	3.1	45.6	34.5	16.8	3.1	40.8	29.4	10.3
MC3	Covelos	4.0	27.9	0.0	11.3	60.7	0.0	30.0	52.5	17.5	0.0	15.6	35.9	16.7
MC4	Pomares	18.2	0.0	0.0	0.6	99.4	0.0	11.0	25.9	63.2	0.0	11.0	25.9	62.4
MC5	Cerdeira	14.0	4.9	0.0	0.0	95.1	1.0	14.3	56.0	28.8	1.0	14.2	56.0	28.1
MC6	Alva	5.8	12.7	0.0	11.5	75.8	1.1	25.6	51.3	22.0	0.3	23.2	43.1	18.6
MC7	Alva	4.0	1.4	0.0	6.4	92.3	0.0	21.2	40.6	38.2	0.0	21.1	38.5	35.9
MC8	Ceira	17.4	0.0	0.0	1.4	98.6	0.0	1.8	40.4	57.8	0.0	1.8	38.7	57.0
MC9	Ceira	3.5	66.3	0.0	0.0	33.7	3.3	17.5	12.6	66.6	0.0	0.0	0.0	0.0
MC10	Mondego	3.8	68.7	25.2	0.0	6.1	65.8	19.5	7.7	7.1	0.0	0.0	0.0	0.0

**Table 2 ijerph-20-00032-t002:** Laboratory analysis techniques; their precision, quantification and detection limits (mg/L); and standards used to determine the major and minor ions.

	Technique	Precision	Q.L.	D.L.	Standard
Br^−^	IC	10%	0.04	0.01	ISO 10304-1 (2007)
Cl^−^	IC	10% (15% <6.0 mg/L)	0.2	0.08	ISO 10304-1 (2007)
NO_3_^−^	IC	10% (15% <3.0 mg/L)	1	0.1	ISO 10304-1 (2007)
PO_4_^3−^	MAS	10% (15% <0.50 mg/L)	0.2	0.1	SMEWW 4500-P B, E
SO_4_^2−^	IC	10% (15% <6.0 mg/L)	2	0.8	ISO 10304-1 (2007)
Ca^2+^	ICP-OES	10%	0.3	0.1	ISO 11885 (2007)
K^−^	ICP-OES	10%	0.1	0.03	ISO 11885 (2007)
Mg^2+^	ICP-OES	10%	0.1	0.03	ISO 11885 (2007)
Na^2+^	ICP-OES	10%	0.3	0.1	ISO 11885 (2007)
Al	ICP-OES	10%	0.01	0.002	ISO 11885 (2007)
As	ICP-MS	10%	0.001	0.0004	ISO 17294-2 (2016)
Ba	ICP-OES	10%	0.002	-	ISO 11885 (2007)
Fe	ICP-OES	5% (10% <20 µg/L)	0.01	0.002	ISO 11885 (2007)
Mn	ICP-OES	5% (7.5% <20 µg/L)	0.01	0.001	ISO 11885 (2007)
Ni	ICP-MS	10%	0.001	0.0001	ISO 17294-2 (2016)
Pb	ICP-MS	10% (15% <10 µg/L)	0.001	0.0001	ISO 17294-2 (2016)
Sr	ICP-OES	10%	0.01	-	ISO 11885 (2007)
Zn	ICP-OES	5% (10% <20 µg/L)	0.01	0.002	ISO 11885 (2007)

IC—Ion Chromatography, MAS—Molecular Absorption Spectroscopy, ICP-OES—Inductively Coupled Plasma Optical Emission Spectrometry, ICP-MS—Inductively Coupled Plasma Mass Spectrometry.

**Table 3 ijerph-20-00032-t003:** Statistical representation of the studied parameters after removal of outlier values.

	Units	N	Min.	Q1	Mean	Q3	Max.
EC	(µS/cm)	89	36.9	65.1	95.4	121.8	289.0
pH	(-)	88	6.0	6.5	6.8	7.0	7.9
DO	(mg/L)	70	2.9	6.0	7.5	9.0	12.7
Turb	(NTU)	75	0.3	3.2	12.3	16.3	91.3
Alk	(mg/L)	78	2.8	6.4	11.4	13.6	32.0
Br^−^	(mg/L)	79	0.01	0.03	0.04	0.05	0.09
Cl^−^	(mg/L)	87	0.08	6.70	10.24	11.20	37.23
HCO_3_^2−^	(mg/L)	90	3.42	8.14	16.78	20.42	57.29
NO_3_^−^	(mg/L)	79	0.15	1.85	3.35	4.60	10.50
PO_4_^3−^	(mg/L)	67	0.05	0.05	0.05	0.05	0.05
SO_4_^2−^	(mg/L)	78	2.10	4.23	6.58	8.00	19.00
Ca^2+^	(mg/L)	88	0.97	2.05	3.53	4.35	12.00
K^−^	(mg/L)	85	0.25	0.67	1.19	1.40	4.00
Mg^2+^	(mg/L)	89	0.78	1.70	2.44	3.00	5.60
Na^2+^	(mg/L)	87	3.80	6.05	8.64	9.80	24.00
Al	(µg/L)	88	1.00	13.00	27.32	35.33	107.00
As	(µg/L)	83	0.20	0.69	1.60	1.70	6.61
Ba	(µg/L)	90	0.53	1.97	4.24	5.87	15.00
Fe	(µg/L)	88	1.00	27.64	62.70	85.58	215.80
Mn	(µg/L)	87	0.50	2.96	12.95	17.23	54.00
Ni	(µg/L)	87	0.05	0.05	0.37	0.50	1.90
Pb	(µg/L)	88	0.05	0.05	0.84	1.28	3.40
Sr	(µg/L)	89	5.00	14.65	20.84	25.75	50.00
Zn	(µg/L)	67	1.00	1.00	6.84	9.81	34.00

N—total number of observations, Min—minimum, Q1—first quartile, Q3—third quartile, Max—maximum.

**Table 4 ijerph-20-00032-t004:** Pearson correlation coefficients between the studied influences and water quality parameters.

	EC	pH	DO	Turb	Alk	Br^−^	Cl^−^	HCO_3_^2−^	NO_3_^−^	PO_4_^3−^	SO_4_^2−^	Ca^2+^
P	−0.12	−0.10	−0.07	0.17	−0.03	−0.19	−0.09	−0.12	−0.08	c	−0.22	−0.14
P5	0.19	−0.18	−0.27 ^a^	0.16	0.27 ^a^	−0.09	−0.01	0.15	0.01	c	−0.04	0.11
P10	−0.09	−0.33 ^b^	0.20	0.39 ^b^	−0.05	−0.05	−0.15	−0.14	0.31 ^b^	c	−0.18	0.03
AvSlp	−0.42 ^b^	−0.19	0.08	−0.09	−0.21	−0.15	−0.43 ^b^	−0.21 ^a^	−0.17	c	−0.31 ^b^	−0.46 ^b^
SClt	0.27 ^a^	0.32 ^b^	−0.01	0.08	0.48 ^b^	0.09	0.12	0.32 ^b^	0.08	c	0.26 ^a^	0.49 ^b^
SCrb	0.14	0.29 ^b^	0.04	0.15	0.46 ^b^	−0.04	−0.02	0.36 ^b^	0.04	c	0.20	0.45 ^b^
Ign	0.43 ^b^	0.04	−0.09	0.00	0.01	0.04	0.46 ^b^	0.17	0.40 ^b^	c	0.33 ^b^	0.35 ^b^
Mtm	−0.58 ^b^	−0.29 ^b^	0.08	−0.08	−0.40 ^b^	−0.09	−0.47 ^b^	−0.42 ^b^	−0.41 ^b^	c	−0.49 ^b^	−0.70 ^b^
Art	0.17	0.30 ^b^	0.04	0.15	0.48 ^b^	−0.04	−0.01	0.37 ^b^	0.07	c	0.22	0.48 ^b^
Agr	0.60 ^b^	0.03	−0.07	−0.07	0.16	0.30 ^b^	0.64 ^b^	0.21 ^a^	0.46 ^b^	c	0.46 ^b^	0.46 ^b^
For	−0.14	−0.39 ^b^	0.07	−0.15	−0.27 ^a^	0.28 ^a^	0.00	−0.27 ^a^	−0.14	c	−0.13	−0.46 ^b^
Shv	−0.38 ^b^	0.03	−0.05	0.01	−0.30 ^b^	−0.35 ^b^	−0.33 ^b^	−0.25 ^a^	−0.20	c	−0.36 ^b^	−0.32 ^b^
AArt	0.44 ^b^	−0.07	0.21	−0.09	0.12	0.17	0.17	0.22 ^a^	0.63 ^b^	c	0.42 ^b^	0.35 ^b^
AAgri	0.25 ^a^	−0.02	0.39 ^b^	0.01	−0.06	0.08	0.11	0.08	0.47 ^b^	c	0.23 ^a^	0.12
AFor	−0.17	−0.17	0.46 ^b^	0.06	−0.22	0.10	−0.19	−0.16	0.09	c	−0.08	−0.35 ^b^
AShv	−0.33 ^b^	0.05	0.38 ^b^	0.14	−0.27 ^a^	−0.18	−0.30 ^b^	−0.15	−0.01	c	−0.22	−0.38 ^b^
	**K^−^**	**Mg^2+^**	**Na^2+^**	**Al**	**As**	**Ba**	**Fe**	**Mn**	**Ni**	**Pb**	**Sr**	**Zn**
P	−0.06	−0.18	−0.19	0.45 ^b^	−0.07	−0.18	0.14	−0.06	0.16	−0.03	−0.17	0.15
P5	0.10	0.00	−0.05	0.10	−0.07	0.06	0.16	0.09	0.04	−0.28 ^b^	0.11	0.09
P10	0.00	−0.11	−0.15	0.47 ^b^	0.07	−0.16	−0.02	0.16	0.12	−0.10	−0.10	−0.06
AvSlp	−0.45 ^b^	−0.26 ^a^	−0.36 ^b^	−0.11	−0.28 ^b^	−0.52 ^b^	−0.32 ^b^	−0.35 ^b^	−0.07	−0.07	−0.35 ^b^	−0.13
ClstS	0.14	0.37 ^b^	0.09	−0.01	−0.08	0.49 ^b^	0.08	0.19	0.08	−0.07	0.34 ^b^	0.05
CarbS	0.12	0.18	−0.01	0.02	0.02	0.36 ^b^	0.00	0.37 ^b^	0.03	−0.02	0.26 ^a^	−0.01
Ign	0.62 ^b^	−0.03	0.51 ^b^	0.10	0.83 ^b^	0.20	0.28 ^b^	0.28 ^b^	−0.02	0.09	0.35 ^b^	−0.01
Mtm	−0.63 ^b^	−0.24 ^a^	−0.50 ^b^	−0.08	−0.62 ^b^	−0.55 ^b^	−0.30 ^b^	−0.43 ^b^	−0.04	−0.03	−0.58 ^b^	−0.03
Art	0.14	0.19	0.00	0.01	0.04	0.38 ^b^	0.02	0.38 ^b^	0.02	−0.02	0.28 ^b^	−0.01
Agr	0.70 ^b^	0.25 ^a^	0.63 ^b^	0.12	0.67 ^b^	0.46 ^b^	0.23 ^a^	0.30 ^b^	0.02	0.03	0.50 ^b^	0.09
For	−0.19	0.05	−0.03	−0.08	−0.27 ^a^	−0.21	−0.24 ^a^	−0.26 ^a^	0.02	−0.03	−0.22 ^a^	−0.02
Shv	−0.34 ^b^	−0.36 ^b^	−0.31 ^b^	−0.02	−0.15	−0.45 ^b^	0.05	−0.30 ^b^	−0.05	0.03	−0.37 ^b^	−0.03
AArt	0.50 ^b^	0.24 ^a^	0.40 ^b^	−0.02	0.53 ^b^	0.26 ^a^	0.06	0.26 ^a^	−0.05	−0.10	0.44 ^b^	0.03
AAgri	0.32 ^b^	0.03	0.26 ^a^	0.19	0.43 ^b^	0.14	0.03	0.20	−0.05	−0.07	0.18	0.07
AFor	−0.23 ^a^	−0.06	−0.14	0.09	−0.18	−0.21 ^a^	−0.26 ^a^	−0.13	−0.06	−0.13	−0.21	−0.02
AShv	−0.28 ^b^	−0.35 ^b^	−0.27 ^a^	0.15	−0.06	−0.39 ^b^	−0.15	−0.13	−0.05	−0.06	−0.32 ^b^	−0.10

^a^. Significance at 0.05 probability level (2-tailed), ^b^. significance at 0.01 probability level (2-tailed). Precipitation (P), precipitation in the last 5 (P5) and 10 days (P10), average slope (AvSlp), percentage of clastic sedimentary (SClt), carbonate sedimentary (SCrb), igneous (Ign) and metamorphic (Mtm) geology, percentage of artificial (Art), agricultural (Agr), forest (For) and shrub/herbaceous vegetation (Shv) and fire-affected area’s equivalents (A).

**Table 5 ijerph-20-00032-t005:** Multiple regression analysis models between natural and anthropogenic factors and the water states of the Mondego River for the studied period.

DV	Regression	R	R2	*p*
EC	EC = 40.296 + 15.005 AArt + 1.85 Agr + 0.536 SClt	0.683	0.467	0.000
pH	pH = 7.269 − 0.01 For − 0.003 P10	0.527	0.278	0.000
DO	DO = 6.938 − 0.054 P5 + 0.052 AFor	0.544	0.296	0.000
Turb	Turb = 7.158 + 0.119 P10	0.388	0.151	0.001
Alk	Alk = 11.590 + 0.294 SCrb + 0.123 P5 − 0.041 Mtm	0.584	0.341	0.000
Br^−^	Br^−^ = 0.023 + 0.000415 Agr + 0.000267 For	0.411	0.169	0.001
Cl^−^	Cl^−^ = 2.925 + 0.349 Agr	0.642	0.412	0.000
HCO_3_^2−^	HCO_3_^2−^ = 21.124 + 0.329 SCrb − 0.092 Mtm	0.463	0.215	0.000
NO_3_^−^	NO_3_^−^ = 3.329 + 1.499 AArt − 0.014 Mtm + 0.008 P10	0.704	0.495	0.000
SO_4_^2−^	SO_4_^2−^ = 8.145 + 1.339 AArt − 0.035 Mtm	0.575	0.331	0.000
Ca^2+^	Ca^2+^ = 1.612 + 0.87 AArt + 0.089 SCrb + 0.046 SClt + 0.027 Ign	0.764	0.584	0.000
K^−^	K^−^ = 0.968 +0.295 AArt + 0.028 Agr − 0.007 Mtm	0.775	0.601	0.000
Mg^2+^	Mg^2+^ = 2.433 + 0.316 AArt + 0.016 SClt − 0.011 Shv	0.524	0.274	0.000
Na^2+^	Na^2+^ = 3.841+ 0.229 Agr	0.626	0.392	0.000
Al	Al = 18.652 + 1.749 P + 0.135 P10	0.568	0.323	0.000
As	As = 1.446 + 0.724 AArt + 0.034 Ign − 0.016 For	0.877	0.769	0.000
Ba	Ba = 6.804 − 0.046 Shv + 0.032 SClt − 0.027 Mtm	0.667	0.445	0.000
Fe	Fe = 123.876 − 3.995 AvSlp − 1.934 Agr + 0.633 Ign	0.430	0.185	0.001
Mn	Mn = 18.687 + 0.466 SCrb − 0.115 Mtm	0.485	0.235	0.000
Pb	Pb = 1.05 − 0.019 P5	0.275	0.076	0.009
Sr	Sr = 22.439 + 4.21 AArt − 0.079 Mtm + 0.078 SClt	0.666	0.443	0.000

Precipitation (P), precipitation in the last 5 (P5) and 10 days (P10), average slope (AvSlp), percentage of clastic sedimentary (SClt), carbonate sedimentary (SCrb), igneous (Ign) and metamorphic (Mtm) geology, percentage of artificial (Art), agricultural (Agr), forest (For) and shrub/herbaceous vegetation (Shv) and fire-affected areas equivalents (A).

## Data Availability

Not applicable.

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
