# Peer review of "The Rural Fires of 2017 and Their Influences on Water Quality: An Assessment of Causes and Effects"

_ijerph, 2022, doi:10.3390/ijerph20010032_

Round 1
Reviewer 1 Report
This manuscript investigated the correlation between rural fire of October 2017 and surface water quality parameters in the catchments. However, it is difficult to find the correlation between the rural fires and water parameter. What is different between the fire-affected areas (MC 1-8) and without fire-affected areas (MC 9-10). It is better to provide the data about the elemental analysis of contaminated air (or ash) from the rural fires. Rural fire and different land-use ways are two mainly factors affecting the water parameters. Besides those non-point source pollution, is there any contribution from point source pollution? Other comments are below,
1. The water quality parameters include not only inorganic (ions and trace elements) but also organic substances. Why is TOC not determined and evaluated in this study?
2. Study area. “MC” should give the full name in the first time.
3. Figure 1. How to get the fire-affected area, please give the explain and source.
4. Table 1. give the meaning of “0.0”
5. Water sampling and analysis. The analysis of Ions, metal and metalloids were conducted by ion chromatography, molecular absorption spectroscopy, inductively coupled plasma optical emission spectrometry and inductively coupled plasma mass spectrometry. It is recommended that what elements were detected by detail equipment and quality control should be provided, such as detection limits, precision.
6. Table 2. please give the meaning of Q1 and Q3, and provide the unit of alkality(Alk). Give the detail elemental species in this table. The std. Dev is meaningless for data from different sampling sites.
7. Results. “it was observable that all values of PO4 were constant” please give the reason of constant concentration of PO4-.
8. Pearson correlation. When these data conform to normal distribution, the results of pearson correlation are meaningful.
9. Table 4. Most of the coefficient (R2) of regression analysis is low 0.5, please give the reasonable explain on the necessity of regression analysis.
10. Discussion. How to difference the effects of agricultural land-use and rural fires on water quality parameters? Please give further the explain and discussion.
11. Writing the correct species of ions and elements in the manuscript, such as NO3-, SO42- etc.
Author Response
Response to Reviewer 1 Comments
Point 1: This manuscript investigated the correlation between rural fire of October 2017 and surface water quality parameters in the catchments. However, it is difficult to find the correlation between the rural fires and water parameter. What is different between the fire-affected areas (MC 1-8) and without fire-affected areas (MC 9-10). It is better to provide the data about the elemental analysis of contaminated air (or ash) from the rural fires. Rural fire and different land-use ways are two mainly factors affecting the water parameters. Besides those non-point source pollution, is there any contribution from point source pollution? Other comments are below,
Response 1: The authors would like to thank the reviewer for the time and work in reviewing, and surely improving the manuscript.
Regarding the difference between fire-affected areas (MC 1-8) and non-fire-affected areas (MC 9-10), although the immediate areas of the MC9 and MC10 watersheds were not directly affected by fire, they are tributaries of the areas affected by previous fires and, therefore, the water can receive, even if very diluted, products from the upstream fire. As for the question regarding airborne contamination, inducing elemental contamination by ash or other particles, this was not an option for investigation. Air quality conditions were not monitored, and further studies with these data are not known, so regional conditions of air dispersion with deposition and dragging to water lines were assumed.
Point 2: The water quality parameters include not only inorganic (ions and trace elements) but also organic substances. Why is TOC not determined and evaluated in this study?
Response 2: While the reasoning behind the reviewer's question is understandable, the authors chose not to spend resources evaluating TOC or DOC. In previous studies, for similar situations, all TOC or DOC values were below the limit of detection. To clarify this option, a reference has now been introduced in the methodology, in the line “The surface water was not analyzed for Dissolved Organic Carbon or Total Organic Car-bon, since in the same conditions all values have been below the detection limits [21]”.
Point 3: Study area. “MC” should give the full name in the first time.
Response 3: We thank the reviewer for noticing this lapse in the manuscript. The first reference to the monitored catchments was added in the second paragraph of the section 2. Study Area, in the line “The monitored catchments (MC) also have high variable morphology, as the southern-most present higher slopes, contrasting to flatter catchments to the north and particularly to the east (Table 1).”.
Point 4: Figure 1. How to get the fire-affected area, please give the explain and source.
Response 4: Once again the authors would like to thank the reviewer for noticing this error in the manuscript. In fact, it was not mention the source of the fire-affected area. In the section 3.1 Measurements of the catchments characteristics, the previous line “Vector-based datasets were collected for altimetry [26], geology [27] and LUO [28].” Was changed to “Vector-based datasets were collected for altimetry [26], geology [27], LUO [28] and the Copernicus Emergency Management Service (EMS) - Mapping (Copernicus).”. Also, to better address how the fire-affected area was determined, on the same section, the line “The LUO was overlapped with the fire-affected areas (A) defining how much of each type of LUO was impacted” was changed to “The LUO was overlapped with the Copernicus EMS – Mapping to define how much each LUO was impacted by fire, defining the fire-affected areas (A)”.
Point 5: Table 1. give the meaning of “0.0”.
Response 5: Despite the description of the table mentions “The AvSlp is in degrees and all areas in percentage”, we do agree with the reviewer that, maybe, it was not very clear what “0.0” corresponded to 0.0% of the watershed, instead we have changed the sentences to “The AvSlp is in degrees and all areas in percentage of the watershed”. Hopefully, in this manner will be clearer.
Point 6: Water sampling and analysis. The analysis of Ions, metal and metalloids were conducted by ion chromatography, molecular absorption spectroscopy, inductively coupled plasma optical emission spectrometry and inductively coupled plasma mass spectrometry. It is recommended that what elements were detected by detail equipment and quality control should be provided, such as detection limits, precision.
Response 6: The authors agree with the reviewer that the original text regarding the analysis of the major and minor ions could be improved in order to become clear all the chemical analysis conducted. To this effect, the authors added a new table, Table 2, describing the laboratory analysis conducted, the standards used, the precision of each method, and quantification and detection limits for all major and minor ions.
Point 7: Table 2. please give the meaning of Q1 and Q3, and provide the unit of alkality(Alk). Give the detail elemental species in this table. The std. Dev is meaningless for data from different sampling sites.
Response 7: We thank the reviewer for noticing some mistakes on this table, as well as the suggestions on improving it. A legend with the meaning of the statistical measures (N, Min., Q1, Q3, Max.) were added. The units for alkalinity, which was missing, has been corrected. The elemental species details were also added. We removed the standard deviation from the table as the information is unrelated for the representation of the results. However, due the need of explaining sample sizes, the N column was added.
Point 8: Results. “it was observable that all values of PO4 were constant” please give the reason of constant concentration of PO4-.
Response 8: The authors understand the reviewer’s concern about this particular sentence as it was not very well explained the reasoning behind a major ion would be constant in 90 samples. The sentence was corrected from “From the preliminary statistical analysis, and after the exclusion of the outliers (Table 3), it was observable that all values of PO43- were constant.” to “From the analytical results, it was noticed that almost all of the PO43- concentrations were below the detection limit, and after the exclusion of its outliers it was observable that all the remaining values of PO43- were bellow detection levels, making this variable constant in all accounted samples.”.
Point 9: Pearson correlation. When these data conform to normal distribution, the results of pearson correlation are meaningful.
Response 9: The Spearman correlation analysis was also tested, providing results similar to the Pearson coefficient. Pearson correlation as the advantage of measuring the covariance between variables rather than ranking the cases. As we previously removed outliers and without the outliers most variables revealed normal distributions, we opted to keep the Pearson correlation.
Note that the Pearson correlation was applied to select the most adequate independent variables, ensuring that the regression analysis was not damped by variables without correlations. To make this clearer, we added in sub-section 3.3 (Statistical analysis) "This procedure was adopted to select the most useful set of independent variables for the regression analysis." We also changed the phrase "The data was evaluated through a Pearson correlation, allowing to define statistically significant relationships ..." to "After removing all outliers, the covariance between variables were evaluated with the Pearson Correlation Coefficient...".
Point 10: Table 4. Most of the coefficient (R2) of regression analysis is low 0.5, please give the reasonable explain on the necessity of regression analysis.
Response 10: The authors understand and agree with the reviewer point that most of the regression analysis were low, as not all explanatory variables were used, like it was previously explained, as well as, there should be more explanatory variables that were not accounted for in this study. In this study, the regression analysis serves the purpose of understanding the which of the selected explanatory variables (rainfall, geology, land-use and rural fires) are the most influential for the studied parameters.
Point 11: Discussion. How to difference the effects of agricultural land-use and rural fires on water quality parameters? Please give further the explain and discussion.
Response 11: Fire affected agricultural areas appear to have specific responses to the wildfires. Arsenic could be likely associated with agriculture, but as it did not pass the tests that preceded the model construction no conclusion can be drawn from the present research regarding this element and its association with agricultural areas. It is demonstrated that other elements, such as Cl- and Na+ are strongly influenced by agricultural activities.
Point 12: Writing the correct species of ions and elements in the manuscript, such as NO3-, SO42- etc.
Response 12: The authors thank the reviewer for noticing this oversight. We have added the charges as per the reviewer request.
Reviewer 2 Report
Thank you for the paper entitled "The rural fires of 2017 and their influences on water quality: an assessment of causes and effects", submitted to the International Journal of Environmental Research and Public Health. Overall I found the paper quite interesting, as relationship between water quality and fires is not very common research topic in the scientific literature. The paper provided some insight into the impact of land cover and fires on stream water chemistry and although the used methods are relatively common, there are some interesting findings. However, I the current form the manuscript should not be published and I recommend some major changes and improvements before publication within Journal of Environmental Research and Public Health. My general remarks are listed below:
· The introduction section should be rewritten to provide more insight into recent research about land effects on stream water quality; in consequence some additional references should be reported. Moreover, the objectives of the paper were not clearly stated; please provide the main objective of the study, as well as specific objectives.
· In the study area section please provide the information about the type of the climate in the study area; the Köppen-Geiger climate classification is recommended to use due to its global recognition. Furthermore, there is no information about streamflow regime of the investigated streams.
· The methods were described very briefly and several questions should be made to the authors to provide more details and explanations. What devices (name of the manufacturer, resolution, precision) were used for electrical conductivity (EC), pH, dissolved oxygen (DO), turbidity (Turb) and alkalinity (Alk) measurements? When the measurements were carried out? How LUO data from 2018 was updated for studied periods? Why the authors did not use the land cover data from Sentinel 2 Global Land Cover, which is pan-European dataset with 10-meter resolution? What CLC classes were used for four computed classes (Art, Agr, For, and Shv)? What weather stations were selected for the precipitation data download; were they representative for the investigated catchments?
I also have doubts about the statistical methods used in the manuscript. Any regression model, should be approximated on the basis of appropriate number of cases (samples); at least 15-20 cases per one explanatory variable are recommended, while in the current study only ten catchments were used. Moreover, why the Pearson correlation coefficient was used for the correlation analysis? This suggests that the statistical distribution of the variables was normal. This is not clear for me and this section must be rewritten; taking into account the previous suggestions will affect the results section
· The description of the results should be more sophisticated; please provide also some graphical visualization of the results (maybe some charts?). Moreover, discussion section is definitely too short and superficial and it should be definitely strengthened with additional, recent references. Finally, in my opinion too little attention has been given to fires and their impact on terrestrial and lotic ecosystems; after all, this is the main problem of the study.
Author Response
Response to Reviewer 2 Comments
Point 1: Thank you for the paper entitled "The rural fires of 2017 and their influences on water quality: an assessment of causes and effects", submitted to the International Journal of Environmental Research and Public Health. Overall I found the paper quite interesting, as relationship between water quality and fires is not very common research topic in the scientific literature. The paper provided some insight into the impact of land cover and fires on stream water chemistry and although the used methods are relatively common, there are some interesting findings. However, I the current form the manuscript should not be published and I recommend some major changes and improvements before publication within Journal of Environmental Research and Public Health. My general remarks are listed below:
Response 1: The authors would like to thank the reviewer for the time and work in reviewing the manuscript, as it assisted in improving it.
Point 2: The introduction section should be rewritten to provide more insight into recent research about land effects on stream water quality; in consequence some additional references should be reported. Moreover, the objectives of the paper were not clearly stated; please provide the main objective of the study, as well as specific objectives.
Response 2: The authors appreciate the reviewer proposal for the introduction. References for more recent land-use studies have been added, despite as it as mentioned, these tend to focus particularly on nitrogen and phosphorous. In the last paragraph of the introduction the main objective has been rewritten.
Point 3: In the study area section please provide the information about the type of the climate in the study area; the Köppen-Geiger climate classification is recommended to use due to its global recognition. Furthermore, there is no information about streamflow regime of the investigated streams.
Response 3: The authors thank the reviewer for these suggestions and have included: a) some information regarding streamflow regime (second line of the first paragraph of the study area section); b) Köppen-Geiger climate classification of the region (last paragraph of the study area section, before Table 1).
Point 4: The methods were described very briefly and several questions should be made to the authors to provide more details and explanations. What devices (name of the manufacturer, resolution, precision) were used for electrical conductivity (EC), pH, dissolved oxygen (DO), turbidity (Turb) and alkalinity (Alk) measurements? When the measurements were carried out? How LUO data from 2018 was updated for studied periods? Why the authors did not use the land cover data from Sentinel 2 Global Land Cover, which is pan-European dataset with 10-meter resolution? What CLC classes were used for four computed classes (Art, Agr, For, and Shv)? What weather stations were selected for the precipitation data download; were they representative for the investigated catchments?
Response 4: The authors thank the reviewer for the advice regarding the methods section, and hopefully all the requesites have been fulfilled in order to totally reproduce this study.
In order to be clear how each parameter were measured, the authors added the line “The waters’ EC, pH, DO were measured using the WTW multiparameter probe 340i and the turbidity using the HANNA HI 93102 Turbidity Portable Meter. To determine the water’s alkalinity, it was used the ISO 9963-1 standard.”, after the line “In each campaign, the waters’ electrical conductivity (EC), pH, dissolved oxygen (DO), turbidity (Turb) and alkalinity (Alk) were determined in situ [22].” that was previously written.
So that would not be any confusion regarding the timing of the measurements the sentence “In the first hydrological year (2017-2018), with exception of February, monthly surface water monitoring campaigns were conducted. These campaigns started in November and conclude by the end of the rainy season, in June. In the second hydrological year (2018-2019), there were two campaigns, one in the rainy season (April) and the other in the dry season (September).” was changed to “The surface water monitoring campaigns were conducted during two hydrological years (2017/18 and 2018/19). During the first hydrological year, 7 campaigns were conducted, from November to June, with a periodicity around the 36 days. While in the hydrological year of 2018/19, it was conducted two campaigns, one in the rainy season (April) and the other in the dry season (September).”.
Also, to address the timing of the laboratory analysis, after the sentence “These analyses were conducted by an accredited laboratory, the Itecons - Instituto de Investigação e Desenvolvimento Tecnológico para a Construção, Energia, Ambiente e Sustentabilidade” it was added the sentence “Analysis of major anions were conducted in a timeframe inferior to 24 hours of the sampling.”.
The authors understand the reviewer’s concern regarding this part of the methodology not being clearly stated in the original version of the manuscript. Therefore, the line “LUO data from 2018 was updated for studied periods, using satellite imagery [29], …” was changed to “The CORINE LUO data which was from 2018 had to be updated for studied periods. Importing the Google Earth CNES/Airbus imagery from the studied catchments to ArcGIS, and each shapefile to intended timeframe [29].”.
Regarding the question that the reviewer posed on why the authors did not use the land cover data from Sentinel 2 Global Land Cover, which has a very high resolution. The authors understand the argument of the reviewer, since it was not very clear which criteria was used to choose the datasets. The sentence “The criteria for choosing the datasets were the availability and resolution proximity between them” was added. Regarding the usage of a higher resolution dataset, the other used datasets were of a lower resolution which would not be balanced in comparison. Since higher resolution geology and soil data for the study area is incomplete, is was opted to use the Corinne Land cover.
Although the CORINE LUO classification has Artificial, Agricultural classes, the authors agree with the reviewer that it might have not been obvious that the forest and shrub/herbaceous vegetation sub-classes were used to compute these areas. Therefore, the line “LUO data from 2018 was updated for studied periods, using satellite imagery [29], and simplified in artificial (Art), agricultural (Agr), forest (For) and shrub/herbaceous vegetation (Shv).” was changed to “The CORINE LUO data which was from 2018 had to be updated for studied periods. Importing the Google Earth CNES/Airbus imagery from the studied catchments to ArcGIS, and each shapefile to intended timeframe [29]. The CORINE LUO shapefiles were combined afterwards, into artificial (Art), agricultural (Agr) areas. Also forest (For) and shrub/herbaceous vegetation (Shv), were created by combining forest and shrub and/or herbaceous vegetation associations sub-classes.”.
Finally, regarding the question posed by the reviewer about from which weather stations was the precipitation data downloaded, and if these station were representative of the studied catchments, the authors appreciate and agree with the reviewer’s suggestions. Therefore, we have changed the sentence “Weather stations close to the catchment were used to acquire the precipitation data.” to “All precipitation data was collected from weather stations, close to the catchments, with continuous precipitation data for the study period (Coimbra: 40º12’0’’N, 8º26’60’’W and Viseu: 40º39’60’’N, 7º54’0’’W).”. The authors also would like to add that the stations used are the closest representative of the studied catchments with continuous records.
Point 5: I also have doubts about the statistical methods used in the manuscript. Any regression model, should be approximated on the basis of appropriate number of cases (samples); at least 15-20 cases per one explanatory variable are recommended, while in the current study only ten catchments were used. Moreover, why the Pearson correlation coefficient was used for the correlation analysis? This suggests that the statistical distribution of the variables was normal. This is not clear for me and this section must be rewritten; taking into account the previous suggestions will affect the results section
Response 5: The authors thank the reviewer for noticing that it was not specifically written in the manuscript the amount of samples used in the regressions analysis. Although only 10 catchments were monitored, they were monitored in 9 campaign, with some fixed and shifting conditions (i.e. precipitation, geology, land use, fire-affected area). Taking this into account, the authors argue that the 10 catchments present somewhat 90 dissimilar situations. After removal of outliers of each parameter, the authors used correlations in order to select which explanatory variables to use in the regression analysis. As an example for Ca2+, the authors removed 2 outliers, from the 90 samples, and correlated these samples to the explanatory variables, ending with a model which only uses 4 explanatory variables (AArt, SCrb, SClt, Ign) as it is show on Table 5. In order to improve the manuscript, the authors added the number of samples used to build each of the regression models to table 3.
As it was not very clear that the Pearson correlation analysis was used to select independent variables to be used in the regression analysis, the line “This procedure was adopted to select the most useful set of independent variables for the regression analysis.” in the 3.3. Statistical analysis sub-section.
Point 6: The description of the results should be more sophisticated; please provide also some graphical visualization of the results (maybe some charts?). Moreover, discussion section is definitely too short and superficial and it should be definitely strengthened with additional, recent references. Finally, in my opinion too little attention has been given to fires and their impact on terrestrial and lotic ecosystems; after all, this is the main problem of the study.
Response 6: Although the authors understand and thank the reviewer’s suggestion on providing a graphical visualization of the results, it was not possible to do this request. While at first, it was attempted to transform the table 3 to a multiple boxplot loses much information, as the variability from all studied elements is very high, leaving some elements with barely visible distribution and others with max values outside of the plot in comparison. Also, the author would prefer to keep the column with the N (cases) which were added in the new version of the manuscript in order to better explain how many samples were used in the model construction. Although, another option could have been the inclusion of the residual plots, which were originally sent in supplementary material. Although, after the reviewer comment, the authors still try to include them in the manuscript, however, due to the high number of plots (21), it was not possible to include them without occupying more than a page or having a low quality image, making the plots unusable. Therefore, the authors decided to keep them in supplementary materials as previously submitted.
The Discussion section has been rewritten, with the addition of recent references, in particular to the sub-section dedicated to the fire, and its influence in surface water chemistry.
Round 2
Reviewer 2 Report
The authors addressed properly most of the comments raised by the Reviewer. The manuscript has been improved and the authors have tried to do their best. However, In my opinion the authors have not provided a comprehensive and satisfactory answer to two remarks. Firstly, although the discussion section was strengthened, in the introduction section I haven't noticed any significant changes. I also suggested to add more that one study objective. Secondly, the authors wrote that: ”the Pearson correlation analysis was used to select independent variables to be used in the regression analysis, the line “This procedure was adopted to select the most useful set of independent variables for the regression analysis.””. Even if the Pearson correlation coefficient was used only for select independent variables, specific assumptions (statistical distribution of data) should always be met. So, in the current form, I recommend the minor revision of the manuscript.
Author Response
Response to Reviewer 2 Comments
Point 1: The authors addressed properly most of the comments raised by the Reviewer. The manuscript has been improved and the authors have tried to do their best. However, In my opinion the authors have not provided a comprehensive and satisfactory answer to two remarks. Firstly, although the discussion section was strengthened, in the introduction section I haven't noticed any significant changes. I also suggested to add more that one study objective. Secondly, the authors wrote that: ”the Pearson correlation analysis was used to select independent variables to be used in the regression analysis, the line “This procedure was adopted to select the most useful set of independent variables for the regression analysis.””. Even if the Pearson correlation coefficient was used only for select independent variables, specific assumptions (statistical distribution of data) should always be met. So, in the current form, I recommend the minor revision of the manuscript.,
Response 1: The authors would like to thank the reviewer for, once again, expend some time reviewing the manuscript. Hopefully, the changes provided answer the rerquests. The introduction was partially rewritten adding new references which establish the theme, previous works and gaps in the literature.
A secondary objective was added, as per request of the reviewer, that captures the latter part of the discussion involving the argument wether the fires impacted the water consumption. To this effect, the line “Moreover, this study aims to discuss whether the founded water parameters, particularly those related to fires, may have an impact in public health, specifically regarding the water’s safety consumption.” was added in the last paragraph of the introduction, after the main objectives.
Regarding the statistic analysis, the authors added lines violations of assumptions for both the Pearson correlation coefficient, as well as the regression analysis were conducted. As well as, the express cases were the assumption could not be met. Concerning the Pearson correlation coeficient, the line “Violations of assumptions of the Pearson correlation coefficient (e.g. level of measurement, related pairs, inexistence of outliers) were assessed beforehand. A violation of assumptions was detected for PO43-.” was added at the end of the first paragraph of the section 3.3 statiscal analysis. While for the regression analysis, the setence “For each tested model, violations of assumptions were assessed beforehand.” was added to the second paragraph of the section 3.3 statistical analysis.